# Towards an Integrated System as Point-of-Care Device for the Optical Detection of Sepsis Biomarkers

**Ambra Giannetti** [1],*, **Cosimo Trono** [1], **Giampiero Porro** [2], **Claudio Domenici** [3], **Mariarita Puntoni** [3] and **Francesco Baldini** [1]

1   Istituto di Fisica Applicata "Nello Carrara", IFAC-CNR, 50019 Sesto Fiorentino (FI), Italy;
    c.trono@ifac.cnr.it (C.T.); f.baldini@ifac.cnr.it (F.B.)
2   DataMED S.r.l., 20068 Peschiera Borromeo (MI), Italy; giampiero.porro@datamedsrl.com
3   Istituto di Fisiologia Clinica, IFC-CNR, 56124 Pisa, Italy; domenici@ifc.cnr.it (C.D.);
    mariaritapuntoni@gmail.com (M.P.)
*   Correspondence: a.giannetti@ifac.cnr.it; Tel.: +39-055-522-6329

**Abstract:** Severe infection and sepsis are a common, expensive, and frequently fatal conditions in critically ill patients. The sepsis diagnosis is not trivial, since it is an extremely complex chain of events involving inflammatory and anti-inflammatory processes, cellular reactions, and circulatory disorders. For these reasons, delay in diagnosis and initiation of drug treatments have shown to be crucial for this pathology. Moreover, a multitude of biomarkers has been proposed, many more than for other pathologies. In order to select optimal treatments for the highly heterogeneous group of sepsis patients and to reduce costs, novel multiplexed tools that better characterize the patient and his or her specific immune response are highly desired. In order to achieve the fundament of drastically improved multi-analyte detection and to attain low limits of detection in diagnostics, the area of point-of-care testing (POCT) technology is developing quickly, leading to the production of instruments, the reliability of which is continuously increasing. For this purpose, a selection of two biomarkers—C-reactive protein (CRP) and neopterin (NP)—was studied in this paper and a fluorescence-based integrated optical system, suitable for future POCT applications, was implemented that is capable of performing the simultaneous measurement of the two different biomarkers in replicate. A limit of detection of 10 and 2.1 $\mu g\ L^{-1}$ was achieved for CRP and NP spiked in commercially available human serum, respectively. Moreover, measurements on both biomarkers were also performed on serum samples collected from septic patients.

**Keywords:** C-reactive protein; neopterin; POCT; sepsis; immunoassay; biochip; fluorescence

## 1. Introduction

Sepsis, defined as the systemic inflammatory response to a confirmed or suspected source of infection, is the most severe infection-related condition, and its identification can be particularly difficult in the initial stages. The microorganisms which can cause sepsis are bacteria, viruses, or fungi; the early stage symptoms are, for example, a temperature above 38 °C or below 36 °C, accelerate cardiac frequency (>90 beats per minute), accelerated respiratory rate (>20 breaths per minute) or $PaCO_2$ less than 32 mm Hg, and also the white blood cell count is altered (more than 12,000 or less than 4000 cells per microliter, or more than 10% immature forms) [1]. This pathology can be described as a three-stage syndrome, starting with sepsis and progressing through severe sepsis to septic shock. The advanced stage of the disease is characterized by organ failure, with symptoms such as significantly decreased urine output, abrupt change in mental status, and decrease in platelet count [2], as observed in patients hospitalized in intensive care units or in patients admitted in emergency departments.

Sepsis is a public health emergency, with increasing incidence, high mortality, and a high health-economic risk, which is often under-evaluated. In the case of less severe situations, the mortality of sepsis is evaluated at around 7% of patients, while it increases to almost 50% in cases of septic shock [3–5]. The key point of an efficient therapy is the appropriate early diagnosis, in order to activate an adequate and effective treatment. However, in many cases, clinical signs are not easily recognizable, since evident symptoms of the pathology are missing and, consequently, the diagnosis is escaping. Even nowadays, clear diagnosis of sepsis is complicated by highly variable symptoms and non-specific criteria, and it is still difficult to be formulated [4]. For this purpose, a specific algorithm—Sequential (sepsis-related) Organ Failure Assessment (SOFA)—has been defined, with the aim of providing a score capable of identifying sepsis among patients who are critically ill with suspected infection [4,6,7]. Notwithstanding this, sepsis is still often diagnosed and treated too late, and an early diagnosis is necessary for the selection of the optimal treatment for this highly heterogeneous group of sepsis patients.

In the framework of this complicated clinical picture, the on-time and simultaneous measurement of those compounds circulating in blood that are overproduced in sepsis—the so-called sepsis biomarkers—can be of extreme help to physicians. Even if single-analyte systems, which have the advantage of simpler procedure and easier measurement setup, have been developed, their application remains challenging due to the lack of specificity of a single biomarker for sepsis [7]. Therefore, the diagnostic can potentiality be increased by the identification of a suitable panel of different biomarkers.

Under these perspectives, the importance of having a POCT platform capable of measuring sepsis biomarkers for a secure early stage diagnosis is clear, but a portable POCT device for sepsis with the US Food and Drug Administration (FDA) approval is still missing [8,9].

In the last years, there has been tremendous research in the area of point-of-care-testing applications for the detection of sepsis [10–12] and the identification of infectious pathogens [13], with the aim to treat sepsis during its early stage, before it becomes more dangerous. Among the proposed systems for the detection of sepsis biomarkers based on optical platforms, Buchegger P. et al. [14] described a protein microarray that simultaneously quantified three interleukins (IL-6, IL-8 and IL-10), tumor necrosis factor alpha (TNF-$\alpha$), S-100, procalcitonin (PCT), E-Selectin, C-reactive protein (CRP), and neopterin (NP). The immunoassays were based on either capture antibodies or derivatives of the analytes, spotted onto functionalized glass slides for their covalent attachment, on top of which the sandwich assay or the binding inhibition assay were performed. In order to have a faster interaction and an increased signal, Dy647 streptavidin conjugated magnetic nanoparticles were added after the incubation step with biotinylated detection antibody. The fluorescence signals were measured by a commercial non-confocal scanner. Using a volume of 4 µL of serum sample diluted 1:10 (40 µL in total), with 2.5 h of incubation time, the following LODs were reached: 1.2 pg mL$^{-1}$ for IL-6, ng mL$^{-1}$ for S-100 0.89, 2.7 ng mL$^{-1}$ for E-Selectin, and 0.13 µg mL$^{-1}$ for CRP. Very recently, Belushkin A. et al. [9] proposed a device based on gold nanoparticles binding to plasmonic gold nanohole array, supporting SPR and localized surface plasmon modes. The nanoplasmonic imager, not yet integrated, can potentially become a portable device, considering the low weight and the small dimensions of the components. Using a volume of 20 µL of serum sample (diluted 50,000 times in the case of CRP), in 15 min of incubation time, the following LODs were reached: 21 pg mL$^{-1}$ for procalcitonin and 36 pg mL$^{-1}$ for CRP.

Among the great number of possible non-specific biomarkers for sepsis (CRP, NP, PCT, TNF-$\alpha$, some interleukins, cytokines, etc.) [15–17], CRP and NP have been selected in this work for a multiple assay development.

In 1983 Pepys et al. [18] described CRP as the first acute-phase protein. Acute-phase response was the name given to a situation in which the concentration of several plasma proteins was altered as a consequence of different form of infection, inflammation, tissue damage, etc. Due to its non-specificity, for many years, CRP has not been considered a useful clinical parameter for the diagnosis of sepsis. Nowadays, it is measured as a sensitive systemic marker for the acute-phase symptoms and sepsis [12,18,19], possibly associated with correlated biomarkers.

As for NP, it is a pteridine derivative produced by activated monocytes/macrophages [20] and it is nowadays considered as a biomarker of cell-mediated immunity. During or after sepsis, elective surgery, and severe trauma, neopterin is released into the plasma [21–23]. As already stated, each single biomarker is not specific for a single pathology, including sepsis. Nevertheless, the levels of both biomarkers, CRP and NP, increase in viral, bacterial, fungi, and parasitic infections, which proves the importance of having a combined measurement in a multi-assay biochip.

This paper describes the implementation on a polymethylmethacrylate (PMMA) biochip of the bioassays for the determination of two markers for sepsis—CRP and NP—inserted in a suitable integrated portable POCT prototype. Calibration curves for both the two biomarkers were achieved and stored in the device and automatic measurements of CRP and NP were performed on serum samples of different septic patients. A total volume of 5 µL of serum sample diluted 1:10 (50 µL in total) was necessary for the immunoassay performances, which give LODs of 10 µg L$^{-1}$ and 2.1 µg L$^{-1}$ for CRP and NP, respectively, within a total assay time of 20 min.

## 2. Materials and Methods

### 2.1. The Multi-Channel Chip

The simultaneous multi-parametric detection was performed with a 13-channel flow-cell, already described [24], with which replicates of the same analyte could be measured. For the sake of completeness, the flow-cell is a PMMA chip, produced by injection molding by the company microfluidic ChipShop (Jena, Germany) on the basis of a design of IFAC and DataMED, and functionalized with -COOH groups by the same microfluidic ChipShop Company. The PMMA chip is the heart of the whole device where the immunoassay for the detection of sepsis biomarkers takes place. It is constituted by two parts: an upper transparent part, properly shaped for optical optimization, and a bottom black part with the grooves for the microchannels through which the analyzed sample flows (Figure 1). The two separated parts are bonded together by solvent bonding, after the chemical functionalization of the upper part with carboxylic groups; both the bonding and the functionalization are proprietary processes of microfluidic ChipShop. The bonding of the two parts defines the thirteen microchannels which are, consequently, characterized by having only one side, the one adjacent to the transparent part, functionalized with the carboxylic group and on which the selective biolayer for the implementation of the immunoassay is realized, as described in the following section. Each microchannel, evidenced in Figure 1 by the red line, is 50 µm high, 600 µm wide, and 9.4 mm long, as shown in the exploded picture. Moreover, the comb-shaped transparent part allows the location of transparent waveguides running over each microchannels, able to collect the fluorescence generated in the biolayer where the immunoassay takes place, as described in the following sections.

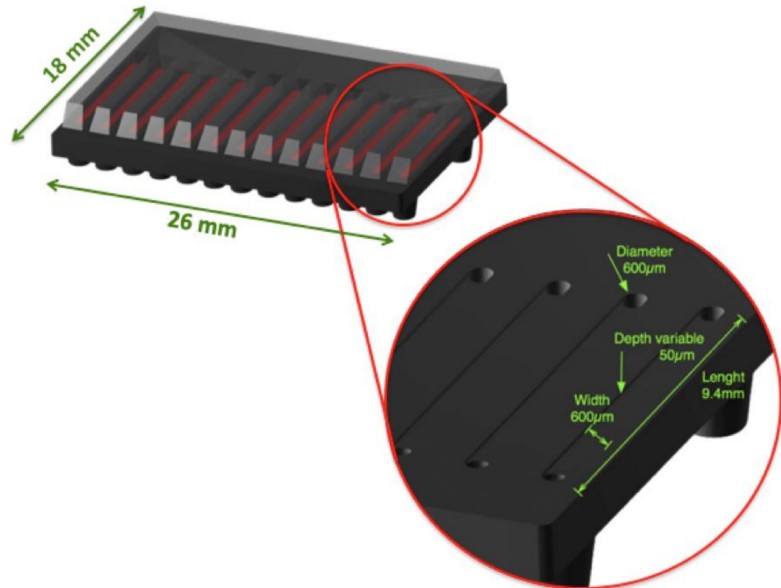

**Figure 1.** The PMMA 13-channel chip. The dimensions of the whole chip are shown (18 × 26 mm). In the exploded picture of the black bottom part, the dimensions of a single microchannel (50 μm high, 600 μm wide, and 9.4 mm long) are indicated. The red lines indicate the location of the microchannels.

## 2.2. The Immunoassay

The selected analytes are CRP and NP. Sandwich assay was implemented for the determination of CRP, while a binding inhibition assay was performed for NP, being this molecule too small for providing two epitopes for the development of a sandwich format. The monoclonal antibodies (mAbs) for CRP, clone C5 as capture mAb, and clone C6 as detection antibody (mAb*), were purchased from EXBIO Praha (Vestec, Czech Republic), whereas the antigen was purchased from Biodesign (Saco, ME, USA). The monoclonal antibodies for NP and the NP conjugated to the BSA were provided by the Veterinary Research Institute of Brno [25]. In particular, the monoclonal antibody produced against neopterin–ovalbumin conjugate by clone 3E2 (mAb 3E2) was selected as the best one among all the tested derivative of NP; moreover, the selected neopterin conjugated was that one with bovine serum albumin (BSA), as reported in [25] and its supplementary materials.

Labeling of the detection antibodies, CRP clone C6 and NP clone 3E2, was carried out by the company EXBIO Praha. Table 1 summarizes the monoclonal antibodies and the conjugated antigen utilized in the two different kinds of assays.

**Table 1.** Combination of capture (mAb) and detection (mAb *) antibodies for CRP and conjugated antigen and detection mAb * for NP. The label of the detection monoclonal antibodies (*) is also indicated.

| Analyte | Capture mAb | Conjugated Antigen | Detection mAb * | Label |
|---------|-------------|--------------------|-----------------|-------|
| CRP | Clone C5 | | Clone C6 | DY647 |
| NP | | NP-BSA | Clone 3E2 | ($\lambda_{exc}$ = 635 nm, $\lambda_{em}$ = 675 nm) |

The COOH-functionalized side of the microchannels was used for the covalent immobilization of the capture antibody for CRP and of the derivative of the antigen for NP; each channel was devoted to the measurement of a single biomarker, being the multiple detection ensured by the presence of the thirteen microchannels. The -COOH groups were activated in flow by means of 2 mM 1-ethyl-3-[3-dimethylaminopropyl] carbodiimide hydrochloride (EDC) and 5 mM N-hydroxy-succinimide (NHS), purchased from Pierce, Rockford, IL, USA. The capture mAb (clone C5 1000 mg L$^{-1}$ for CRP) was covalently immobilized, as well as the NP–BSA conjugated antigen

(1000 mg L$^{-1}$). After 5 min of washing with the phosphate buffered saline (40 mM PBS, pH 7.4, disodiumhydrogen phosphate (Na$_2$HPO$_4$) purchased from Sigma-Aldrich S.r.l., Milan, Italy), the surface was blocked with 0.1% of bovine serum albumin (BSA Sigma-Aldrich S.r.l., Milan, Italy) in PBS for 15 min and finally washed for 5 min with PBST (PBS with 0.05% Tween 20, Sigma-Aldrich S.r.l., Milan, Italy).

The chips were then used for two different purposes: determination of the calibration curves (both in buffer and in commercial human serum for CRP and in commercial human serum for NP) and final measurements on real patient samples.

For the determination of the calibration curves in serum, the two antigens (CRP and NP) were spiked in diluted (1:10) CRP-free serum (HyTest Ltd., Turku, Finland) at different concentrations, and the obtained solutions were pre-incubated out of the chip, with the related detection antibodies, clone C6 and clone 3E2 for CRP and NP, respectively, both for 10 min, with a concentration of 1 mg L$^{-1}$. The dilution of the serum was performed in LowCross buffer/PBS (Candor Bioscience, Wangen, Germany). In particular, the proportion ratio of the serum/LowCross buffer/PBS solution was 1:2:7 in order to reach the final dilution of the serum 1:10 (v/v), considering that the PBS contains the detection antibodies so as to achieve the desired concentration. The total sample volume used was 50 µL.

As for the measurement on human samples, the same procedure was followed with the patient serum samples collected by the hospital. That is to say, dilution 1:10 (serum/LowCross buffer/PBS) and 10 min of incubation with the detection antibodies at a concentration of 1 mg L$^{-1}$. The resulted mixture (50 µL) was allowed to flow into the microchannel chip for incubation with the bio-activated chip surface.

For comparison purposes, the determination of the CRP and NP values in human samples was also achieved by means of standard laboratory procedures:

- CRP was quantified by rate nephelometry (BN ProSpec, Siemens Healthcare Diagnostics, Italy);
- NP was measured by using a commercial ELISA kit (DRG International Inc., Mountainside, NJ, USA).

All experiments were performed in compliance with the relevant laws and institutional guidelines, and the experimentation with human samples was approved by the Ethical Committee of IFC-CNR of Pisa, Italy (Study 2271/07; Protocol CARE-03 WP10).

### 2.3. The Integrated Optical Detection System

The working principle of the device was already described in previous papers [26,27], but it is here shortly presented again for clarity. The optical beam of a laser diode emitting at 635 nm (Hitachi HL6314MG), properly filtered with a band pass filter at 635 nm and focused by means of a cylindrical lens, illuminates from the top of the chip a single microchannel, and consequently the sensing layer. The emitted fluorescence, which comes from the sensing layer where the specific biologic interaction takes place, is mainly coupled to the transparent waveguides running over each microchannel; this occurs since the emitted fluorescence is anisotropic and directed towards the denser medium, i.e., the PMMA, being the fluorophore located at a distance from the medium interface smaller than the emission wavelength; the emitted fluorescence is guided within the PMMA waveguides since the anisotropic emitted fluorescence has well-defined preferential directions, characterized by angles larger than the total reflection angle [11,27–29]. The guided fluorescence is laterally collected by a single plastic optical fiber connected to a cooled photodiode (Hamamatsu S9295). A long-pass edge filter at 647 nm (Semrock LP02-647RU-25) is placed in front of the photodiode, to cut the contribution of the scattered light at 635 nm coming from the source. A motorized translation stage (Thorlabs MTS50X) allows the serial automatic scanning of the 13 different microchannels, with the laser diode, which illuminates the microchannel, and the optical fiber, which collects the fluorescence, moved together from one microchannel to the other. This completely avoids the possible interference coming from adjacent channels, since every channel is interrogated in different times.

## *2.4. The Software*

As graphical user interface, a single software program with two modules was developed by DataMED, to control the fluidics and excitation–detection modules. This approach allows a fast debug and adaptation to any processing and measurement needs. The first module is used to control the movement of the excitation source and to process the fluorescence signal, while the second one is used to control and program the fluidic tasks. With this software, it is possible to define a calibration procedure, to calculate and store the calibration curve and perform an automatic measurement.

## 3. Results and Discussion

### *3.1. The Integrated POCT-Prototype*

In Figure 2, the main units integrated into the POCT-prototype are shown. In particular, Figure 2a shows the render of the optical components and how they are interfaced. At the top, the flow-cell positioned in correspondence of the laser diode emitting at 635 nm is visible. The optical fiber for the collection of the emitted fluorescence collection is connected to the photodiode and moved together with the laser diode (a zoom out of this part is in Figure 2a at the bottom-right). Figure 2b shows the drawing of the fluidic control unit. The developed software allows the creation of fluidic procedures that can be stored and used when necessary. Each procedure can be divided into three separate phases, which make it possible to control the switching of the valve, the activation of the pump, and how long the pump has to work. It is possible to control the delay between each phase and also to repeat the same phase. The integrated system, composed by the optical unit and the fluidic control unit, is embedded into a portable box, as shown in Figure 2c, and it is controlled by a software totally developed by DataMED.

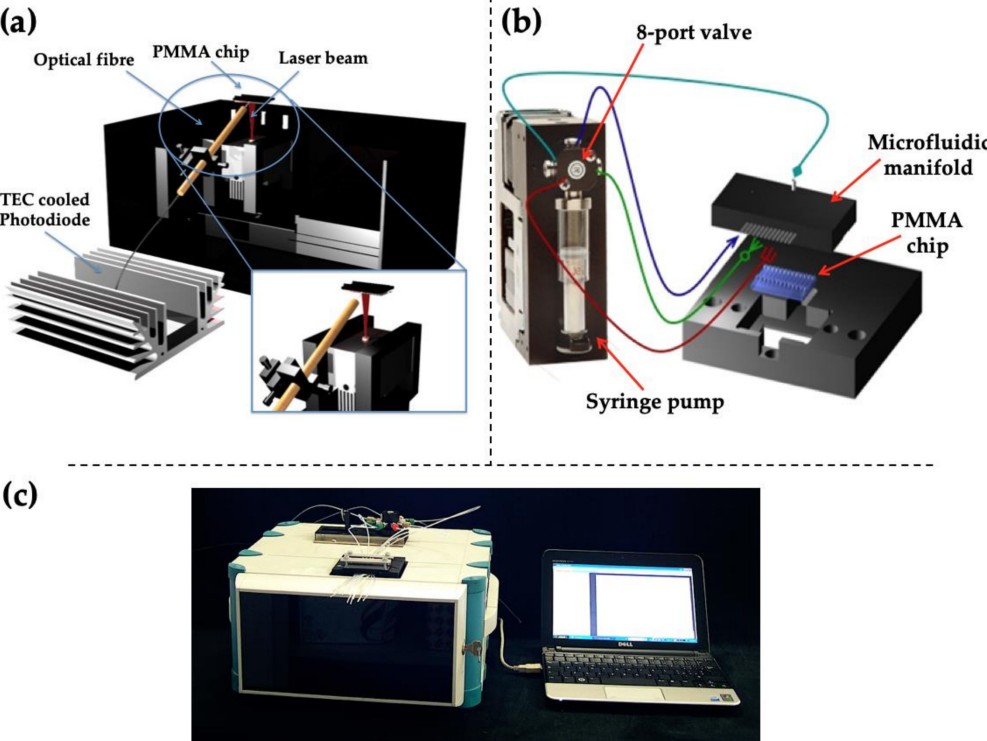

**Figure 2.** The POCT-prototype. (**a**) A render of the optical unit is presented, showing the laser beam as source, the flow-cell interrogated by the optical fiber, and the cooling system. (**b**) The fluidic unit is schematically represented, and it shows the fluidic connection among the PMMA-chip, the microfluidic manifold, and the 8-port valve and the syringe pump. (**c**) Picture of the integrated instrument connected to a laptop.

With respect to the previous published results, the system was optimized with an improved temperature stabilization and a better mechanical stability. Suitable housing for the laser source and for the photodiode was implemented, connected with Peltier cells, which drastically reduced the temperature fluctuations. Figure 3 shows the photodiode detection module, with the preamplified photodiode, the dissipator, and the fan, suitably mounted over the TEC and fan control box. The motorized translation system, constituted by the translation stage, the laser and the mechanical holders of the fiber, was redesigned to provide better mechanical stability and mounted in order to place the stage not directly under the chip, but laterally, with respect to it. This upgrade led to a better reproducibility in the micromovements and solved the problem of the seizing failure induced by fluidic leakages on the translation stage.

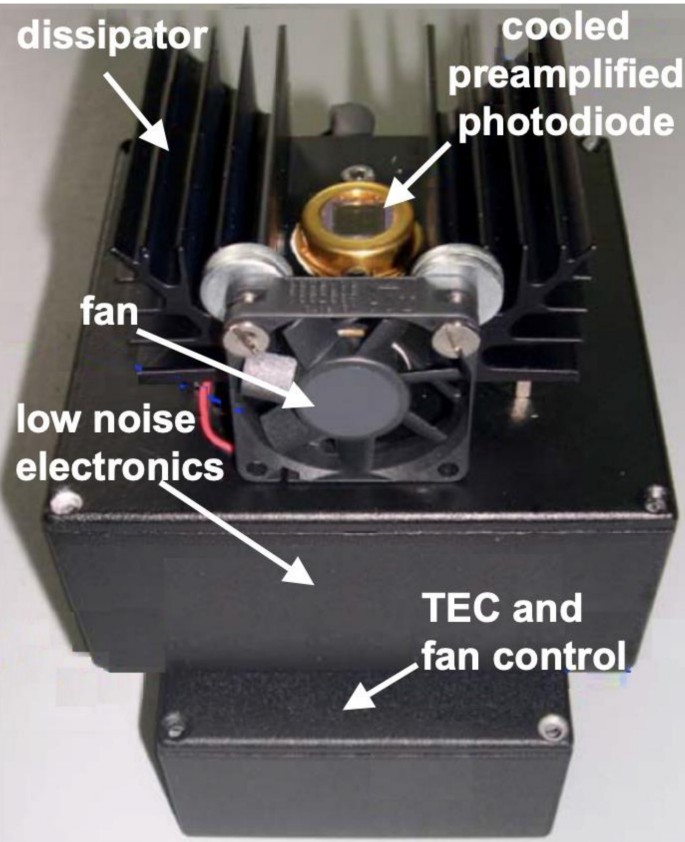

**Figure 3.** The improved photodiode detection module.

Particular attention was devoted to the development of an easy procedure for chip replacement. A stainless-steel manifold was manufactured by DataMED, with one inlet and thirteen outlets connected to the Tygon tubes (internal diameter of 0.5 mm, Gilson Italia S.r.l, Milan, Italy), on which the chip is positioned (Figure 4, left); by exerting a suitable pressure, the chip is connected to the fluidics without any sample leakage. The manifold is equipped with an electrical heater for thermal stabilization of the flow-cell to the desired temperature. A second manifold is fabricated (Figure 4, right) for the realization of the sensing layer (see Section 2.2), with thirteen inlets and thirteen outlets; in this way it is possible to treat each microchannel with different antibodies and realize on each microchannel a different sensing layer; as a matter of fact, three sensing layers for each different biomarker are realized in order to have a triplicate for each biomarker on the same chip.

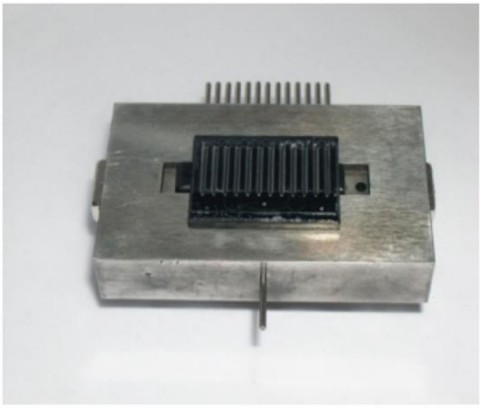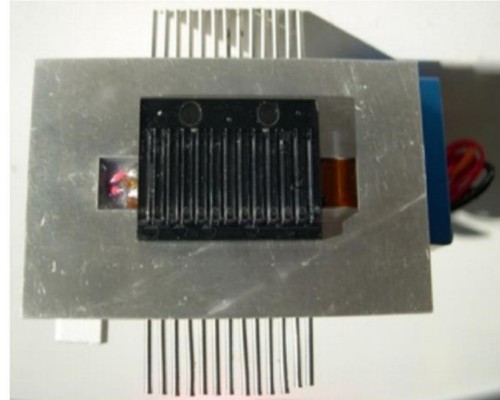

**Figure 4.** The microfluid chip coupled with the manifold. **Left**: the manifold used for the sample measurement with one inlet and thirteen outlets; **Right**: the manifold used for the preparation of the sensing layer with thirteen inlets and thirteen outlets.

As for fluidics, a syringe pump (model PDS4, Hamilton, Switzerland) connected to an 8-port valve (Hamilton, Switzerland) is coupled via Tygon tubing to the microfluidic chip. Figure 5 shows the fluidic scheme used for the immunoassay, where the syringe pump draws the sample from a vial and pushes it inside the chip via the inlet of the manifold, which is also shown in Figure 4, on the left. A vial for washing steps is also present. In the case of the realization of the selective biolayers on the surface of the microchannels, different vials can be connected via Tygon tubing to different microchannels; thanks to the 8-port valve, and using the 13*13 manifold (Figure 4, right), it is possible to draw different solutions in seven different microchannels (the eighth valve is used for waste) with a single procedure; as a matter of fact, it is possible to realize the selective biolayers in all the thirteen microchannels of the chips in only two different steps.

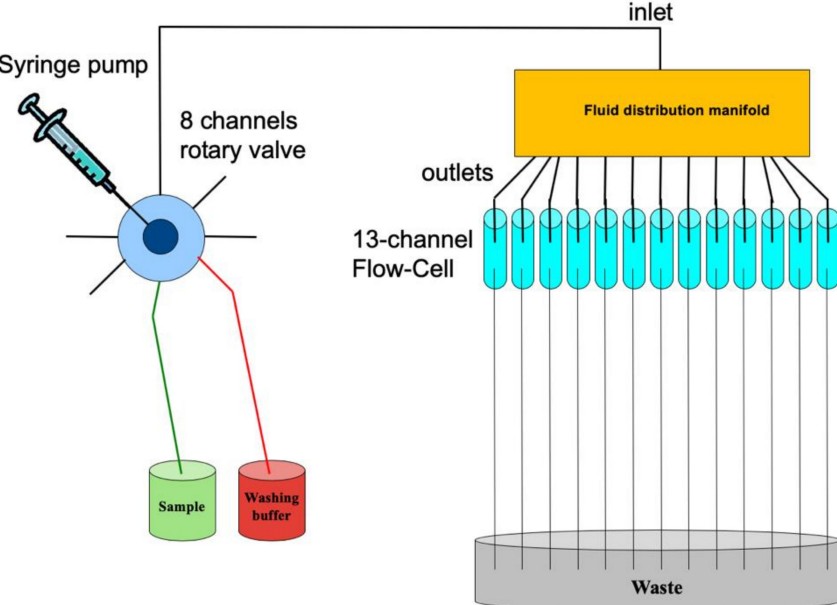

**Figure 5.** The cartoon shows the fluidic scheme used for the immunoassay, where the syringe pump draws the sample from a vial and pushes it inside the chip via the inlet of the manifold.

A flow rate of 10 μL/min is used in the immunoassay; considering that the volume of a single microchannel is roughly 0.28 μL, the total volume within the chip is 3.7 μL. Therefore, a volume of 50 μL is sufficient to run one measurement, and this ensures a very low consumption of the necessary antibodies, an aspect not to be underestimated, considering their high cost.

### 3.2. The Calibration Curves for CRP and NP

The logistic function used for the fitting of the calibration data is here reported:

$$y = F_{max} + \frac{F_{min} - F_{max}}{1 + (x/x_0)^p} \tag{1}$$

where $F_{max}$ and $F_{min}$ are the asymptotes of the sigmoidal curve, $x_0$ is the value of the concentration for which the fluorescence signal is equal to the 50% of the dynamic range, and p is a coefficient which is related to the slope of the curve for $x = x_0$.

In Figure 6, the calibration curves for CRP sandwich immunoassays in PBS buffer and in serum are reported [24]. In particular, the calibration curves were obtained by spiking different concentrations, from 5 µg L$^{-1}$ up to 100 mg L$^{-1}$, of the protein into the PBS buffer (Figure 6a), or from 0.1 µg L$^{-1}$ up to 10 mg L$^{-1}$, in 1:10-diluted serum (Figure 4, right), respectively. As demonstrated in previous works [11,30], the use of LowCross buffer significantly improved the LOD performances of the immunoassay, reducing the non-specific interaction; for this reason, it has been used for all the measurements performed in serum (both commercial and real samples). In the case of the calibration of Figure 6a, each point is the average of ten different values acquired every 10 s on the same microchannel, whereas, in Figure 6b, the reported fluorescence signals were calculated for each concentration of the CRP, taking the average value of the signals acquired with three different channels. In the first case, the standard deviation is very small, practically not visible in the drawn graph, testifying for the good stability of the optoelectronic module; in the second graph, the standard deviation is greater, because it is the average of the measurement performed on three different microchannels, and, consequently, it takes into account the chemical (e.g., differences in the immobilized biolayers) and microfluidic (e.g., small differences in the channel dimensions) fluctuations which clearly affect the reproducibility of the measurement.

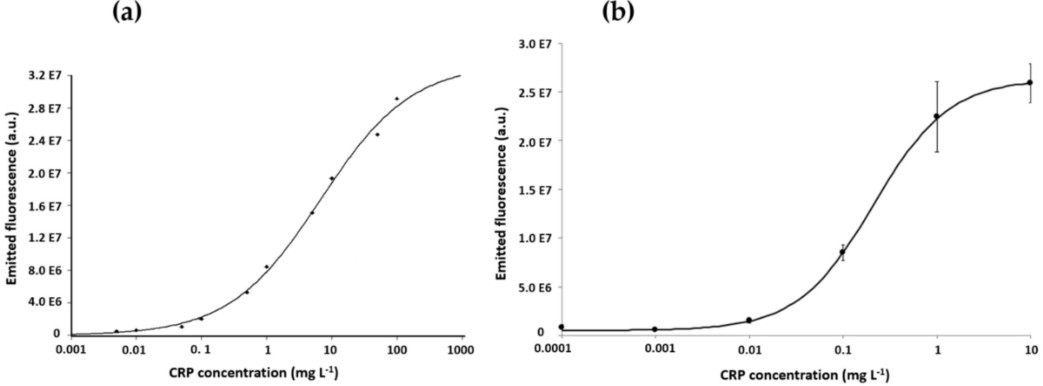

**Figure 6.** (**a**) Calibration curve for CRP (from 5 µg L$^{-1}$ up to 100 mg L$^{-1}$) in PBS buffer; (**b**) calibration curve for CRP (from 0.1 µg L$^{-1}$ up to 10 mg L$^{-1}$) spiked in 1:10-diluted serum (modified from figures in [24]).

The limit of detection (LOD), calculated following the IUPAC rules, as three times the standard deviation of the blank was 0.004 mg L$^{-1}$ and 0.01 mg L$^{-1}$ for CRP in buffer and in serum, respectively.

In Figure 7, the calibration curve for binding inhibition assay related to NP is reported. In particular, the calibration curve for NP was obtained by spiking different concentrations of the protein in 1:10-diluted human serum, from 0.01 up to 1000 µg L$^{-1}$. As for the calibration curve in serum, the dilution buffer was the LowCross buffer, as done for CRP. The reached LOD was 2.1 µg L$^{-1}$.

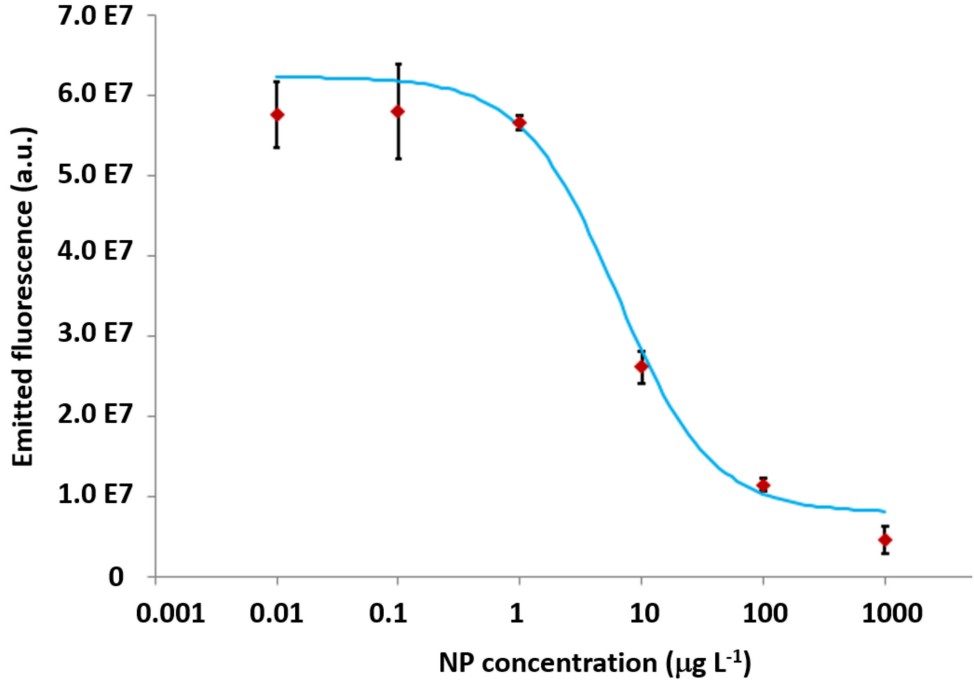

**Figure 7.** Calibration curve for binding inhibition assay for NP (from 0.01 up to 1000 μg L$^{-1}$) spiked in human serum.

Considering the clinical intervals of interest of the two biomarkers, reported in Table 2, it is evident that the dilution of 1:10 of the samples is necessary to achieve the appropriate working range of the calibration curves for both CRP and NP.

**Table 2.** Clinical range of interest for CRP and NP.

|  | Healthy Volunteers | Patients with Sepsis | References |
| --- | --- | --- | --- |
| *CRP (mg L$^{-1}$)* | 1–10 | 20–500 | [31–33] |
| *NP (μg L$^{-1}$)* | 1–5 | 24–105 | [16,34–38] |

### 3.3. The Measurements on Human Samples

Six patient samples were provided by the hospital of the Institute of Clinical Physiology, IFC-CNR, of Pisa (Italy).

In Table 3, the levels of the biomarkers measured with standard technique in the hospital are reported, in comparison with the values obtained with the POCT device described in this paper. Since the measurements were performed in human samples diluted 1:10, the calculated concentrations obtained with the POCT prototype reported in the table are again multiplied by 10, for an easier comparison of the obtained values with the actual concentrations measured with standard methods. The reported measured values, were calculated on the basis of the average of triplicate of the sample measured in three different channels of the same chip. It was possible to measure the NP levels only on two of them.

**Table 3.** Levels of CRP and NP in six septic patients, measured in hospital and with the POCT-prototype described in this paper.

| Patient Identifying N. | CRP (mg L$^{-1}$) Actual Concentration (Standard Method) | CRP (mg L$^{-1}$) Calculated Concentration (POCT Prototype) | NP (µg L$^{-1}$) Actual Concentration (Standard Method) | NP (µg L$^{-1}$) Calculated Concentration (POCT Prototype) |
|---|---|---|---|---|
| 1 | 3.65 | 1.59 | not available | |
| 2 | <0.168 | 0.15 | not available | |
| 3 | 1.34 | 1.34 | not available | |
| 4 | 43.7 | 45.89 | not available | |
| 5 | 15.9 | 10.5 | 12.4 | 13.32 |
| 6 | 0.614 | 0.27 | 6.58 | 5.60 |

Figure 8 shows, for CRP, the correlation graph of the values achieved with the two methods, which demonstrates the validity of the measurements performed with the developed POCT prototype (Pearson correlation coefficient r = 0.991). Considering also the good correlation of the two NP values measured with the POCT prototype with the values achieved with the standard ELISA kit, the possibility to have a point of care device for multiple measurements in about 20 min—a reasonable time to give the possibility to the physicians to elaborate a fast diagnosis—has been demonstrated.

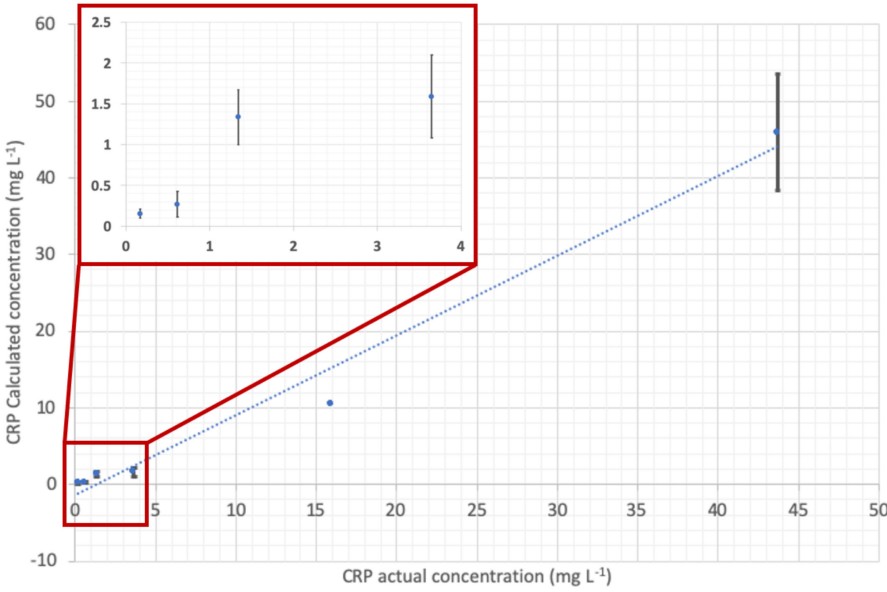

**Figure 8.** Correlation between CRP actual concentration measured in the hospital and CRP calculated concentration measured with the POCT prototype. A zoom-out of the first four measurements is reported in the red square, for a better reading of the data.

## 4. Conclusions

In this paper, a POCT prototype was described with the microfluidics, the optics, and the optoelectronics integrated within a portable device connected with a laptop for data processing by means of a dedicated software. The designed optical biochip, with thirteen different microchannels, allows the multiple detection of biomarkers with the same sample. It is important to observe that, thanks to the dedicated software and the suitable design of the microfluidics, it was possible to carry out on the same chip with the same sample the two different formats at the same time, the sandwich assay for CRP and the binding inhibition assay for NP. Calibration curves or both biomarkers in commercially available human serum were realized, reaching LODs of 10 and 2.1 µg L$^{-1}$, respectively, within a total assay time of 20 min. The calibration curves for both the two biomarkers were then stored in the software device, and automatic measurements of CRP and NP were performed on human serum samples of different septic patients. The good correlation between the biomarker actual concentration

measured in the hospital and biomarker calculated concentration measured with the POCT-prototype showed the reliability of the prototype in the market for POCT medical devices.

**Author Contributions:** Conceptualization, A.G. and C.T.; methodology, A.G. and C.T.; software, G.P.; validation, A.G. and C.T.; formal analysis, A.G. and C.T.; investigation, A.G.; resources, All; data curation, A.G.; writing—original draft preparation, A.G. and C.T.; writing—review and editing, All; visualization, A.G. and C.T.; supervision, F.B.; project administration, F.B.; funding acquisition, F.B. All authors have read and agree to the published version of the manuscript.

**Funding:** This research was partially funded by the European Community and the Tuscany Region within the framework of the FASPEC project (Horizon 2020—PhotonicSensing ERA-NET COFUND; G. A. 688735) and by the 'Health-CARE by biosensor Measurement and Networking' (CARE-MAN) (NMP4-CT-2006-017333).

**Acknowledgments:** The authors wish to thank EXBIO Praha, Czech Republic, for labeling the detection antibodies, and microfluidic ChipShop Jena, Germany, for the realization of the chips and their functionalization. Moreover, the authors wish to thank Milan Franek and Ivo Cernoch, from the Veterinary Research Institute of Brno, for the development of the antibody for neopterin and the derivative of neopterin itself (BSA-neopterin).

**Conflicts of Interest:** The authors declare no conflicts of interest.

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
