# Peer review of "Towards an Integrated System as Point-of-Care Device for the Optical Detection of Sepsis Biomarkers"

_chemosensors, doi:10.3390/chemosensors8010012_

Round 1
Reviewer 1 Report
The authors present a point-of-care device for the multiplexed detection of two biomarkers related with sepsis diagnosis – C-reactive protein and neopterin.
In my opinion there are serious issues to be addressed before to accept the presented manuscript for publication. Major queries are listed below.
In the manuscript the authors stress the need for POC platforms to address the simultaneous detection of the various described biomarkers in early stage. However, despite the acknowledgement of the existence of some competitive alternative technologies in a multiplexed POC format, no description or insight of the performance of such technologies is made in the manuscript.
In the introduction a better insight on the state-of-the-art is required. Relevant references of review and original research papers are missing (e.g. Lab Chip, 2019,19, 728-737; Buchegger et al., Sensors 2012, 12(2), 1494-1508).
Pag 2, line 62-64: Detail in the description of the existing technologies, mentioning the performance, figures of merit and how it compares to the technology being presented, is missing.
In materials and methods are missing important details to understand and replicate the described experiments.
Pag 2, line 94: The bonding of the two microfluidic parts is not explained.
Pag 3, line 95: The functionalization is not explained or supported by a reference. Besides, there are contradictory affirmations regarding the functionalized part of the channel; line 95 - bottom part; line 97 - top transparent part. Figure 1 should help to clarify by pointing out this location.
Is this functionalization method a proprietary process from ChipShop? If yes it should be stated in the manuscript.
Where are the two different biorecognition elements immobilized? It is not clear if the two assays (for CRP and NP) occur in the same channel or at different channels.
Pag 4, line 143: The volume of assayed sample is not stated.
In the Results and Discussion section lack results to support the following affirmations:
Pag 6, line 209: improved mechanical stability (…)
Pag 6, line 211: improved temperature stabilization (…)
Figure 4 – Both plots were already published as it is in other publication, which is not referred in the figure caption. The authors claim that standard deviations of the averaged values are reported (line 242), but neither error bars are presented in Fig 4a) nor any statement is made in the caption of the figure.
In Figure 6 is missing the equation and the correlation factor to support the affirmation of “good correlation” between the two methods.
Reviewer 2 Report
The authors utilized an integrated polymethylmetacrylate (PMMA) POCT biochip for the quantification of two sepsis biomarkers(CRP and NP). Calibration curves for both the two biomarkers were achieved and automatic measurements of CRP and NP were performed on clinical samples. Some specific comments are listed as following.
This work is suffering from the grammar errors. The authors need to carefully check their manuscript. The scheme figure needs to be greatly improved for a better understanding of the immunoassay process inside microfluidic channels. In figure 4, the maximum detectable concentration of CRP in the diluted serum seems to be 10 mg/L. However, in Table 3, the highest detected concentration of CRP is 45.89 mg/L. The data provided by authors is of doubtful reality. It is afraid that the proposed biosensor cannot be applied in real clinical use. The references are out of date and need to be updated to the latest literature.
Reviewer 3 Report
The paper “Towards an integrated system as Point-of-Care device
3 for the optical detection of sepsis biomarkers” reporter the optical system to detect C-reactive protein (CRP) and Neoptorin (NP) moieties in order to prevent sepsis state human patients. The results showed in this work are not a new contribution, because the same author published the same optical system in 2010 [ref. 18 and 19].
For other side the author said that the detected signal was the photoluminescence of the probes molecules used in this system, however the detection phase of the system was building only with fiber optics as collector and single photodiode without using any monocromator. In this sense the collected signal also could have contribution from excitation laser light since any roughness in the surface or elastic deformation of surface (PMMA) could excite the guiding light in the transparent part of the system.
The paper doesn’t reported any information about the selectivity of the proposed system and it didn´t make any experimental assay in relation to study the signal interference coming from the neighborhood channels.
Reviewer 4 Report
This manuscript reports very interesting devices with remarkable limits of detection for CRP and NP. Thus, the manuscript deserves publication on Chemosensors, but after the following issues are addressed.
Major comments.
1) In the introduction, it is clear the importance of detecting CRP and NP, but there is no comparison with other biosensors in terms of limit of detection and speed of measurement. A table containing such a comparison should be added.
2) Line 107-110. The sandwich protocol is useful for this type of measurement and the authors state that they are able to perform measurements with sandwich protocol for CPR and not for NP because the latter is too small in dimension and they prefer to use the inhibition assay. Did the authors try to use polyclonal antibodies instead of monoclonal antibodies? The recognition of different epitopes by polyclonal antibodies should make easier also the detection of NP with sandwich protocol and it is often possible also to increase the limit of detection. In spite of a possible lack of specificity, such an approach may lead to some advantages.
3) In the section “Results and discussion”, the authors describe the POCT-prototype where we can find an 8-valve syringe pump, tygon tubes and so on. Some more details should be added such as the flow-rate, the volumes involved in the whole circuit and, in particular, in the PMMA-chip. An assessment about the cost should be reported since monoclonal antibodies are expensive and small volumes are highly desired.
4) What is the diameter of the tygon tubes? The serum is diluted 1:10; is it enough to avoid the obstruction in the tubes due to the high concentration of the compounds of the serum?
5) In the caption of figure 4b, it is written that the calibration curve starts from 1 µg/L, but in the dose-response curve it starts from 100 ng/L. Please check.
6) Why is the error bar so high for 1 mg/L?
7) Where does the error bar come from? Is it from multiple measurements?
8) What is the sensitivity of the biosensors?
9) The validation curve showed in figure 6 is very interesting, but the figure is unclear. Since 4 out of the 6 points are in the range 0-5 mg/L, it would be appropriate to have a zoom in that region to better see the correlation among the measurements performed by standard method and POCT prototype.
Minor comments.
1) The resolution of figure 4 is not good. In addition, the caption is on the top of the figure rather at the bottom.
2) The error bars in figure 4a are not showed (or, at least, not visible).
3) The resolution of the figure 5 is poor and maybe the colors are not in agreement with the standard of the journal.
4) The style for the citations n.11, 16, 18, 19, 22 and 23 should be revised.
5) The label, font and size in figure 4 in (a) and (b) are different.
Round 2
Reviewer 1 Report
The authors have improved the manuscript according to the reviewers comments. Figures were significantly improved, methods better described and results better presented and discussed. However the English language and style still need to be carefully revised.
A short comment on the Figure 6 and 7 axis legend. The units in the y-axis should be standardized in terms of decimal places and format.
Author Response
We thank the reviewer for this further effort on improving the quality of the manuscript.
The legend of Y-axes of the figures 6, 7 and 8 have been updated in a uniformed format and the English language was carefully revised (changed parts are in blue).
Reviewer 2 Report
The manuscript has been significantly improved and I now warrants publication in Chemosensors.
Author Response
Thank you.
Reviewer 3 Report
all questions were answered satisfactorily.
Author Response
Thank you.
Reviewer 4 Report
The authors revised the manuscript addressing all the comments raised by the Reviewers. In its current form the manuscript is suitable for publication on Chemosensors.
Author Response
Thank you.